# Analysis of Parkinson’s Disease Using an Imbalanced-Speech Dataset by Employing Decision Tree Ensemble Methods

**DOI:** 10.3390/diagnostics12123000

**Published:** 2022-11-30

**Authors:** Omar Barukab, Amir Ahmad, Tabrej Khan, Mujeeb Rahiman Thayyil Kunhumuhammed

**Affiliations:** 1Department of Information Technology, Faculty of Computing and Information Technology in Rabigh (FCITR), King Abdulaziz University, Jeddah 21589, Saudi Arabia; 2College of Information Technology, United Arab Emirates University, Al Ain P.O. Box 15551, United Arab Emirates; 3Department of Information Systems, Faculty of Computing and Information Technology in Rabigh (FCITR), King Abdulaziz University, Jeddah 21589, Saudi Arabia; 4Department of Computer Science, Faculty of Computing and Information Technology in Rabigh (FCITR), King Abdulaziz University, Jeddah 21589, Saudi Arabia

**Keywords:** PD, ensembles decision tree, feature selection, classification, imbalanced class, lasso, information gain, predictors

## Abstract

Parkinson’s disease (PD) currently affects approximately 10 million people worldwide. The detection of PD positive subjects is vital in terms of disease prognostics, diagnostics, management and treatment. Different types of early symptoms, such as speech impairment and changes in writing, are associated with Parkinson disease. To classify potential patients of PD, many researchers used machine learning algorithms in various datasets related to this disease. In our research, we study the dataset of the PD vocal impairment feature, which is an imbalanced dataset. We propose comparative performance evaluation using various decision tree ensemble methods, with or without oversampling techniques. In addition, we compare the performance of classifiers with different sizes of ensembles and various ratios of the minority class and the majority class with oversampling and undersampling. Finally, we combine feature selection with best-performing ensemble classifiers. The result shows that AdaBoost, random forest, and decision tree developed for the RUSBoost imbalanced dataset perform well in performance metrics such as precision, recall, F1-score, area under the receiver operating characteristic curve (AUROC) and the geometric mean. Further, feature selection methods, namely lasso and information gain, were used to screen the 10 best features using the best ensemble classifiers. AdaBoost with information gain feature selection method is the best performing ensemble method with an F1-score of 0.903.

## 1. Introduction

Parkinson’s disease (PD) is a progressive neurological disorder with symptoms ranging from trouble with movement to dementia, autonomic dysfunction, depression, and visual hallucinations [1]. Between seven and 10 million people worldwide are affected currently by PD. The disease affects anywhere from 41 persons per 100,000 in their forties to more than 1900 people per 100,000 in their eighties and beyond. The disease’s incidence, or the rate at which new cases are identified, normally increases with age, but it can be stable in adults over the age of 80. Only about 4% of people diagnosed with PD are under 50 years old. PD affects 1.5 times more men than women [2].

PD significantly impairs patients’ quality of life, family relationships, and social functions, and imposes considerable economic costs on both individuals and societies [3,4,5].

PD is often diagnosed by a physician based on the patient’s complaints and the neurological examination that follows the disease history [6]. Although certain expensive methods such as CT scan, X-ray imaging, dopamine transporter scan, Single Photon Emission Computerized Tomography (SPECT), and others are available, these techniques are useful at detecting PD only if disease is widespread across the brain [7,8]. There are a few unique computer-aided tests that may be used to diagnose PD, such as the finger touch and handwriting laboratory tests [9,10,11,12,13,14]. In recent years, establishing speech tests has emerged as a potential study topic that analyzes phonation and vocal signals in order to classify people with PD. Over 90% of PD patients show a unique pattern of atrophy and speech problems, which is one of the early signs of early-stage PD [15,16,17]. 

The early detection of PD can have a big impact on the disease’s progression and the patient’s quality of life. Machine learning, together with knowledge detection from medical libraries, has long been touted as a possible technique for addressing early diagnostic and prediction problems. Machine learning, as the term implies, is the ability of software to automatically learn and extract meaningful representations from data. A variety of data sets have been used to train machine learning models. The data modalities, which include handwritten patterns, that can be used to aid in the diagnosis of PD [18,19] and are: based on movement-based features [20,21,22]; neuro image-based datasets [23,24,25]; voice-based features [26,27] ; the cerebrospinal fluid (CSF) dataset [28,29]; cardiac scintigraphy [30]; serum-based datasets [30]; and the optical coherence tomography (OCT) dataset [31]. Machine learning enables the integration of data from multiple modalities such as magnetic resonance imaging (MRI) and single-photon emission computed tomography (SPECT) in the diagnosis of PD [32,33]. While past research indicated that recognizable speech signals develop about 84 months following PD onset, recent prospective investigations reveal that objective, accurate assessments can identify speech patterns. [34]. Detecting PD significantly improves when non-invasive, easily accessible, and low-cost patient-generated speech data is used. The present work uses a publicly available dataset that contains speech features extracted from 188 patients, and 64 healthy controls, using a variety of speech signal processing techniques [35]. The dataset used in earlier studies had a problem of class imbalance, where records belonging to people with disease are greater or less than the records belonging to people who are healthy. The unbalanced dataset does not help the machine learning model perform successfully. Various machine learning methods and applications have been developed to handle the class imbalance problem for these datasets [23,24]. Specific model evaluation metrics must be used to evaluate models built using imbalanced datasets. This is true especially for medical datasets, where the target class is in minority. Therefore, in such cases the false negatives are penalized excessively more than the false positives to prevent the model from committing type 1 and type II errors, respectively [25]. An ensemble classifier has been applied to solve various classification problems [26,27,28]. A decision tree classifier is one of the most popularly used machine learning algorithms to build simple interpretable models for decision-making [6]. Certain machine learning algorithms namely, bagged decision trees [31], eXtreme gradient boosting (XGBoost) [30], and random forest [29], are some of the most popular examples of a decision tree ensemble. The ensemble of a decision tree is quite robust and accurate in selecting parameters from small to large dimensional datasets. Decision tree ensemble models have been used to learn from imbalanced data and perform effective classification tasks [32,33]. Various PD speech datasets are imbalanced datasets. Studies have shown that ensemble methods can produce very efficient prediction when the data is properly preprocessed and oversampled [36]. Not many researchers [36,37,38,39,40] have analyzed the data by considering it as imbalanced. In this paper, we have performed an extensive study on an imbalanced dataset using the algorithms developed for imbalanced data. Following are the contributions of our present work.

We considered an imbalanced dataset and performed automatic classification between PD patients and healthy controls to evaluate the robustness of different ensemble methods for class imbalance.Decision tree ensembles have been shown to have excellent performance in different domains. In this study, we carried out extensive performance evaluations of different types of decision tree ensembles such as RUSBoost, isolation forset, RUSBagging, balanced bagging etc.; developed for imbalanced data. To the best of our knowledge, this has never been used by other researcher in this area.We carried out the feature selection using lasso and the information gain method, to achieve the best set of features.

There are different approaches to analyze an imbalanced data set, including deep learning, machine learning, and time series analysis. However, in this paper, we concentrate only on decision tree ensembles. Ensemble methods have shown very good result and very robust result. As per literature, ensemble methods were used to explore imbalanced dataset in various domain. However, these ensemble methods have not been applied to PD dataset. Therefore, we intend to extensively explore the decision tree ensembles such as balanced random forest (RUS), the synthetic minority oversampling technique (SMOTE), bagging, RUS bagging, SMOTEBoost, RUSBoost, and so forth, to predict the PD in its earlier stages. Moreover, various oversampling and undersampling techniques will be used to address the issue of class imbalance in the current PD speech dataset. In a classification task using imbalanced data, accuracy alone is insufficient to compare performance for various individual classifiers and ensembles of classifiers. Therefore, in our study, geometric mean, the area under the precision-recall curve, the area under the receiver operating characteristic curve, sensitivity (SN), specificity (SP), precision, and F1-score performance measures, were used to evaluate and compare the performance for various decision tree-based ensemble models. The methodology in our study to compare the performance of ensemble classifiers and selection of the subset of features is shown in Figure 1.

The rest of the paper is organized as follows: Section 2 examines several comparable research works, as well as their relationship with the current work. The following section discusses the methodology that has been proposed. The next section describes the speech dataset that was used in this study. Then, Section 4 discusses the experimental findings. Section 5 concludes the paper.

## 2. Related Work

The recent research has demonstrated that abnormalities in speech can be utilized as a quantifiable indicator for the early detection of PD [41]. In the early stages of PD, most people experience vocal problems. Voice signals are oscillatory signals that are non-linear and non-stationary. Because around 90% of patients have voice abnormalities early in the disease’s progression, these symptoms can be helpful in detecting the disease [42]. In this section, we discuss the research works that have been carried out to predict PD using a speech signal dataset that is collected at the neurology department of Istanbul University [35].

Sakar et al. [35] compared the effectiveness of Tunable Q-factor Wavelet Transform (TQWT) with the recent feature extraction methods used in diagnosis of PD from voice data. Their study revealed that TQWT has a higher frequency resolution than the classical wavelet transforms and the TQWT feature-based model is more effective than the existing models.

I. Nissar et al. [43] detected Parkinson’s disease using several machine learning algorithms. The proposed model used two types of feature selection techniques, namely, RFE (Recursive Feature Elimination) and mRMR. Different machine learning models have been used and investigated for PD detection, including logistic regression, naive bayes, KNN, random forest, decision tree, SVM, MLP, and XGBoost. Of these, XGBoost with mRMR feature selection technique achieved highest accuracy on all feature subsets, outperforming all state-of-the-art methods.

Yücelbaş [44] proposed the Simple Logistic Hybrid System (SLGS) as a new technique that employs feature analysis to identify PD by gender. The proposed system’s performance was evaluated using a range of statistical metrics. 

Lavalle [45] et al. proposed a method for speech-based PD detection that involves selecting feature subsets and applying four different classifiers to voice recordings from five datasets (gender-based, balanced, and unbalanced) obtained from the largest publicly available dataset for voice-based PD detection. One of the contributions is a performance and complexity improvement over previous work on voice-based PD detection using the same dataset. In the gender-based dataset, the highest detection performance achieved in the female dataset.

Gunduz [46] suggested a classification method for individual voice recordings based on vocal features collected from audio, as well as a hybrid feature reduction approach for extracting robust features. The approach was developed using Variational-Auto Encoders (VAE), and feature selection models were employed. The Fisher-Score and Relief were used as filter-based techniques due to their efficiency in dealing with data with noise, while VAE was selected as a feature extractor due to its ability to retain the latent space’s normal properties during feature creation. Compared to the results obtained without dimension reduction, the results of the proposed model were of increased accuracy. 

Mohammadi [42] et al. used speech data sets to build PD models. They effectively predicted PD using vocal features from the Machine Learning Repository database, which has a limited sample size and imbalanced features. Their technique uses autoencoder training and the encoder section to extract nonlinear information. Stacking the models produced more accurate predictions. The result of their study shows that the use of autoencoders as feature extractors may be beneficial when the total samples are smaller than the number of features, especially when the input is imbalanced. Experiments show that traditional classification models outperform deep learning techniques.

Ashour et al. [47] proposed a unique two-stage feature selection framework for identifying PD patients with voice loss. An SVM classifier was used to calculate the suggested weighted hybrid selected features. The suggested approach employs a cubic kernel-SVM with best performance in detecting voice loss in PD.

Yücelbaşn et al. [48] created the Information Gain Algorithm-based K-Nearest Neighbors (IGKNN) diagnostic system to accurately identify PD using speech data. The dataset was received from UCI, and the characteristics were derived from 252 people’s speech signals. This approach identified Tunable Q-factor Wavelet Transform (TQWT) datasets. This approach was applied to TQWT feature data sets. To obtain the result, the 12 sub feature datasets that make up the TQWT feature sets were evaluated separately, then the sub feature dataset with the best performance was added to the IGKNN model. The IGKNN system’s performance was compared to other studies using the same dataset, and the method presented in this paper outperformed all earlier methods.

By including relevance feature weighting into the Gaussian Mixture Model (GMM), Bchir [49] suggested a novel method for handling the problem of higher dimensionality. To compare the obtained results to the GMM findings and to the most current research in the literature, the GMM with a relevance feature weights technique is employed. The findings demonstrated the usefulness of the suggested technique, produced best accuracy. 

Suvita et al. [50] proposed binary Rao PD classification techniques. The suggested method uses Rao algorithms, which do not require parameter tuning. A hyperbolic tangent V-shaped transfer function converted continuous Rao techniques to binary. Binary Rao is also used to optimize KNN’s ‘k’ value. The recommended methods were evaluated using PD data sets. Using the Friedman rank test, the suggested method’s relevance was determined. The suggested binary Rao algorithms were compared against state-of-the-art algorithms.

Younis Thanoun and Yaseen [51] proposed two types of ensembles learning approaches for PD detection via machine learning: stacking classifiers and voting classifiers. Subsequently, they compared the outcomes of the two approaches. The Stacking Classifier approach outperformed the voting classifier.

Gemci and Ibrikci [52] employed a Feed-Forward Neural Network (FFNN) built with Python’s Keras in the experiment. In epoch 30, the deep learning algorithm successfully classified the PD data set with best accuracy. 

Prasad et.al. [53] predicted PD using a set of 753 vocal features and a two-step classification framework. They proposed a novel technique for PD classification by employing multiple one-way ANOVA on independent vocal features. They proposed to classify PDs using an XGBoost classifier trained on the extracted data. The proposed framework achieves best classification accuracy. 

An effective feature vector extraction pipeline, devised by Xiong and Lu [54], consisted of an Adaptive Grey Wolf Optimization Algorithm and a sparse auto encoder neural network that classified the PD samples from the healthy with high accuracy. The speech dataset was used to evaluate how well the adaptive WSO-sparse auto encoder-LDA model performed in comparison to the benchmarked algorithms.

Schellhas et al. [55] created a technique called Distance Correlation Sure Independence Screening (DC-SIS), which employs a correlation measure to identify features that are most dependent on the response. On PD vocal diagnosis data, this method produces statistically indistinguishable results 90 times faster than the method of mRMR selection.

The research discussed above are not addressed the imbalanced nature of the dataset [35]. The imbalanced dataset will not help the machine learning model perform successfully. Many academics have overlooked this problem, but it has an impact on the classification system’s performance. Oversampling and undersampling approaches can be used to tackle this problem. In some research, the Synthetic Minority Oversampling Technique is utilized to address this problem. Minority samples in SMOTE are created artificially from existing dataset examples. The research that addressed the dataset’s imbalance nature are discussed in the next section.

### Related Literature Which Addresses the Imbalance Problem

Polat [36] proposed a method for detecting an imbalanced class distribution in a PD dataset by combining SMOTE and random forest classifiers. The RF classifier built using balanced data achieved high accuracy. Jain et al. [56] employed deep neural networks with SMOTE oversampling for the prediction problem and achieved best performance in PD detection. 

Lamba et al. [57] proposed a hybrid PD diagnosis system based on speech signals to aid in the disease’s early detection. They used numerous feature selection methodologies and classification algorithms and created the model that produced the best results. The best performance was achieved by combining a random forest classifier and a genetic algorithm.

Hoq et al. [37] created a pair of hybrid models for predicting PD patients using voice data that include the SVM, PCA, and Sparse Auto Encoder (SAE), wherein PCA was used to extract voice feature main components. The secondary model uses a deep neural network of a Sparse Auto Encoder with L1 regularization to compress voice features. Both models sent reduced features to SVM, which categorized the data by learning hyperplanes and projecting it to a higher dimension. The SAE-SVM model outperformed not only the previous PCA-SVM model, but also other standard models such as MLP, KNN, XGBoost and random forest.

Pramanik et al. [38] investigated two newly developed decision forest algorithms: the Systematically Developed Forest and the Decision Forest by Penalizing-Attributes. The Forest by Penalizing Attributes (ForestPA) proved to be a promising PD detector with a minimum number of decision trees and a highest detection accuracy.

We note that not many researchers have used the state-of-the-art algorithm for oversampling and undersampling data and decision tree ensemble for PD detection. Therefore, in the present study, it will be interesting to conduct PD prediction studies by applying algorithms that have been developed for imbalanced datasets.

## 3. Materials and Methods

This section will discuss the PD dataset speech vocal signal, decision tree, ensemble decision tree, features selection methods, and imbalanced evaluation metrics included in the current study.

### 3.1. Parkinson’s Diseases Speech Vocal Dataset

The PD speech signal dataset is collected at the Department of Neurology in Cerrahpasa Faculty of Medicine, Istanbul University, Istanbul, Turkey. The PD speech signal data used in the present study was obtained from the UCI machine learning repository [35]. The PD dataset consists of 188 patients (81 Females and 107 males) with PD, and the rest, 64 samples, were healthy subjects (41 females and 23 males). The ratio of the majority class to the minority class was 2.93, which makes the data imbalanced. In addition, the age of healthy subjects varied between 41 to 82 years, whereas the age of PD patients ranged from 33 and 87 years. All throughout the data collection, the microphone’s frequency response was set to be centered at 44.1 kHz.

All participants were asked to perform the sustained phonation of the vowel “ah” three times. Speech disorders are an earlier sign for PD patients; therefore, speech characteristics are essential in evaluating the PD. The PD speech dataset comprises 753 features (752 speech features + 1 gender) related to speech characteristics [39], presented in Table 1. In the next section, we will discuss classifier ensembles.

### 3.2. Decision Tree Classifier

The decision tree classifier is based on the recursive partition method, where the sample points are split based on a specified split criterion. The C4.5 algorithm employs the information gain ratio as a splitting principle from top to down, reducing the bias towards multivalued attributes [40]. In addition to the information gain ratio, various other splitting criteria have been proposed for the decision tree. The base classifier for this study is the C4.5 classifier (J48) for every ensemble of classifiers.

On the other hand, an ensemble is a method that combines several base classifiers to produce a best classifier for better prediction and stability by reducing individual classifier errors such as variance, noise, and bias [58]. The PD speech signal dataset was imbalanced, and therefore we used various decision tree-based ensembles built for imbalanced class datasets.

### 3.3. Decision Tree Ensembles

This study uses some general ensemble methods, namely AdaBoost [59] and bagging [60]. These ensembles can also combine with any other classifier. In addition, we use some specific type of a decision tree, namely that of a random forest (RF) [61]. Finally, XGBoost [62] implements scalable gradient tree boosting ensemble methods used to enhance the speed and performance of the ensemble model.

### 3.4. Decision Tree Ensembles for Imbalanced Datasets

There are many methods to handle imbalanced datasets. Oversampling and undersampling are two essential techniques to reduce the class imbalance problem. Oversampling is associated with the minority class and undersampling is associated with the majority class [63,64]. Synthetic Minority Oversampling Technique [65] and Random Undersampling (RUS) [63,64] are two important techniques for undersampling and oversampling. Here, RUS is the undersampling method that selects the sample points from the majority class (more instances) associated with the minority class (less instances). Similarly, SMOTE is the oversampling method which increases the minority class in different ratios. In oversampling data generated through SMOTE technique with k = 5 (default) nearest neighbor. SMOTE select random data from the minority class, then select k-nearest neighbors from the data. Artificially data would then be generated between the random data and the randomly selected k-nearest neighbor. For categorical features smote use different interpolation method such as selects the most common class of the nearest neighbors or different distance metric instead of euclidean distance in the encoded space [65]. The RUSBoost algorithm [66] performs the random undersampling with a boosting algorithm. While the RUSBagging [67] performs random undersampling technique using the bagging algorithm. Likewise, the SMOTEBoost [68] algorithm uses SMOTE oversampling with boosting, and SMOTEBagging [69] employs SMOTE oversampling with bagging. The balanced random forest [70] uses random undersampling of the majority class with random forest to imbalanced data. Moreover, balanced bagging combines bagging with random undersampling of the majority class. This study used packages such as weka [71], Imblern [67], and XGBoost [72] for performing various experiments. The experimental setup mentioned in Section 3 and all experimental results are presented in Section 4.

### 3.5. Feature Selection Methods

We have used two feature selection methods: Information gain [73] and Least Absolute Shrinkage and Selection Operator [74,75].

#### 3.5.1. Feature Selection Using Information Gain (IG)

Information gain is the most popular feature selection method in bioinformatics that uses a filtering approach to select relatively essential features [76]. In Weka, feature selection is achieved by a combination of an attribute evaluator and a search method. In this study, we have applied InfoGainAttributeEval with the ranker search method for feature selection. InfoGainAttributeEval provides the information of features with respect to a class. The ranker search method provides rank to features based on attributes. In addition, we evaluate the performance of the set of attributes extracted from various feature selection methods using different ensembling techniques, namely AdaBoost, random forest, and RUSBoost. 

#### 3.5.2. Least Absolute Shrinkage and Selection Operator (Lasso) or L1 Regularization

Lasso is an embedded feature selection method. Lasso selects the important feature from the dataset by reducing the less important feature’s coefficient to precisely zero or to some negative values. The current study also applied the lasso [74] feature selection method on the Parkinson Dataset to find the best subset of features. We use L1 penalty function optimization using a 10-fold cross validation (CV) to select the best subset of features for model building. In addition, measures for the performance of various ensembles, namely AdaBoost, random forest, and RUSBoost, were evaluated using multiple evaluation metrics. Sixteen best features were obtained using the lasso feature selection method. The list of features obtained using lasso is presented in Section 4. 

### 3.6. Evaluation Metrics

Different metrics are used to evaluate classifiers’ performance, such as accuracy, precision, recall, Geometric Mean, SN, SP, and F1-score [77]. However, for the class imbalance problem, accuracy is not enough to evaluate the classifier. Therefore, in addition, recall, G-mean, precision, and F1-score were used for model evaluation. Precision, Recall, and F1-score are discussed in detail in this section. In the current study, we use metrics, namely the Geometric Mean, the area under the precision-recall curve, the AUROC, SN, and SP, to evaluate the performance of the decision tree-based ensemble models [78].

#### 3.6.1. The Area under the Receiver Operating Characteristic (ROC) Curve

The AUROC curve is plotted between TPR (SN/Recall) and false positives rate (FPR = 1 − SP) at various decision thresholds. The AUROC determines the classification model quality and differentiates the two classes in the dataset. The range of AUROC [79] varies from 0 to 1. The baseline random classifier has an AUC value of 0.5. 

#### 3.6.2. Area under the Precision-Recall (PR) Curve (AUPRC)

The AUPRC is an important metric to evaluate the performance of classifiers using an imbalanced dataset. The AUPRC curve is plotted between precision and recall at various decision thresholds. Moreover, AUPRC [80] has no baseline value. The AUPRC value ranges from 0 to 1, where 0 denotes a worst-performing model while 1 represents a best-performing model. 

#### 3.6.3. Geometric Mean (G-Mean)

G-Mean is a standard metric to compute the performance of a model built using an imbalanced dataset. The G-mean [81] is a square root of the product of SN and SP. Maximizing the G-Mean gives an optimal classification boundary particularly for an imbalanced dataset. The range of the G-mean extends between 0 and 1, where 0 indicates the worst performing model and 1 represents the best performing model. In this experiment, we use the G-Mean to evaluate the performance of decision tree ensembles. 

#### 3.6.4. Sensitivity

SN is one of the most popular metrics to evaluate the classifier’s performance trained and tested on an imbalanced dataset [82]. It is the ratio of true positives to the sum of true positives and false negatives. The maximum value of SN is 1 and minimum value is 0. In this study, to calculate the SN, we use the Equation (1).
(1)$$SN=\frac{TP}{TP\;+\;FN}$$

#### 3.6.5. Specificity

It is the ratio of true negatives to the sum of true negatives and false positives. SP is an important metric to evaluate the performance of model built using an imbalanced dataset. The maximum value of SP is 1 which indicates the best model whereas 0 represents the worst model [83,84,85]. The SP of the model is calculated using Equation (2), as shown below: (2)$$SP=\frac{TN}{TN\;\;+\;\;FP}$$

## 4. Experimental Setup, Results, and Discussion

This section will represent the experimental setup and the results of our investigation.

### 4.1. Experimental Setup and Software Packages

Experiments were carried out using Weka tools and the Python (scikit-learn) programming language. In this study, we used the Imblern and XGBoost packages. The default parameters of all the classifiers were used to train and test different predictive models. We used 10-fold cross-validation using participant and we ensured that each participant’s recordings were either in the training set or in the test set and a fixed ensemble size of 50 was used for all the experiments. Our study used standard decision tree ensembles and some already developed ensemble classifiers tabulated in Table 2 with corresponding software packages. This experiment oversampled the minority class, and we utilized the SMOTE technique using Weka. Similarly, for random undersampling, we used SpreadsubSample supervised filter (Weka) and the Imblearn (python) package. We implemented RUS undersampling using the Weka and Imblearn packages with the majority class to change the ratio of the majority class to the minority class.

### 4.2. Comparative Study of Various Decision Tree Ensemble Models Built Using the Imbalanced Dataset

Different types of decision tree ensembles were compared by using various performance measures. The results are tabulated in Table 3. For example, the decision tree ensemble built using imbalanced datasets performs best in SP. However, standard classifiers performance was the best in terms of Accuracy, AUROC, and SN. Random forest [RF] is best for two performance measures: AUROC and SN. RUSBoost measures best performance in SP. AdaBoost perform best in Accuracy. Similarly, a Balanced random forest is best in SP. The AUROC, SN, and SP are broadly used performance measures for imbalanced datasets. The AUROC for random forest is 0.952, the SN 0.983, and the SP 0.622. This means that the classifier is very confident about the correctly classified samples (high AUROC), but it diagnoses many of the HC as PD patients (low SP) while RUSBoost performed best for SP with the values of 0.792 respectively. This means that the RUSBoost classifier is very confident about diagnoses many of the HC (high SP). By contrast, the balanced random forest also provided the best SP with the value 0.792. The study shows that decision tree ensembles for imbalanced class problems obtained better results for this Parkinson speech dataset.

### 4.3. Decision Tree Ensembles Using Various Sampling Techniques

In this study, we used different sampling techniques such as SMOTE and RUS. Sampling is a technique to overcome imbalanced class datasets [86]. We used for our study the performance metrics AUROC, SN, SP, G-Mean, and AUPRC, which measure imbalanced datasets performance. In Table 4 and Table 5, the outcomes of the SMOTE oversampling method are tabulated. In this experiment, we used various ratios of majority class data points and minority class ones. The experimental results on the actual data points are tabulated in Table 4 and Table 5. The outcomes recommend that the best G-Mean was achieved as 0.87 by AdaBoost with a ratio of 0.75 datasets, whereas the best SN was achieved as 0.983 by RF with the actual dataset. Similarly, the best SP was acquired as 0.784 by AdaBoost and random forest.

In addition, the best AUPRC was achieved as 0.988 by random forest with the proportion of 0.50 dataset, whereas the best AUROC was achieved as 0.963 by random forest with the ratio of 0.50 dataset. Thus, the AUROC and AUPRC results were improved with bagging and AdaBoost concerning oversampling except for a single decision tree. Similarly, in terms of G-mean, AdaBoost and RF improved the outcome for oversampling except for the single decision tree. On the other hand, the SMOTE oversampling method negatively affected the performance of various ensembles. The poor performance of SMOTE oversampling process is due to the presence of noisy minority class points.

In the case of undersampling, the AdaBoost achieved the best outcomes in terms of AUROC and AUPRC. The best AUPRC was 0.986 with the data with a proportion (ratio) of 0.75, whereas the best AUPOC was 0.954. Random forest achieved the best outcome for the original data, whereas AdaBoost achieved the best outcome for the data with a proportion of 0.5. The best outcomes (SN, SP, and G-Mean) were obtained with random forest and bagging. Results with the RUS undersampling method are presented in Table 6 and Table 7. The best outcomes of SP and G-mean were 0.892 and 0.87, respectively, with the data with a ratio of one (1) whereas the best SN was 0.939 with the data with a proportion of 0.50. Similarly, random forest, single decision tree, and AdaBoost achieved best with the original dataset. The outcomes revealed that all the ensembles have no same type of effect due to the sampling method. In contrast, the best outcomes among all the ensembles (decision tree classifiers) were gained with the AdaBoost and bagging for the data with a ratio of 0.75 formed with random undersampling.

### 4.4. Ensemble Size Effect in Imbalanced Datasets

Ensemble is a combination of the different classifiers. Size of ensemble means several classifiers. In the current experiment, we carried out different ensemble sizes such as 20, 50, 100 and 200. The performance metrics AUROC, AUPRC, SN, SP, and G-mean are included in our study. The results, presented in Table 8 and Table 9, suggest that performance is slightly improving or remains constant with ensemble size. For example, random forest (AUROC and AUPRC) is best, with an ensemble size of 200. Similarly, SN was best with bagging, and random forest with ensemble sizes of 200 and 50. SP is best with a balanced random forest with an ensemble size of 100, and G-Mean is best with RUSBoost with 200.

### 4.5. Feature Selection and Comparative Performance Evaluation of Best Ensemble Classifiers with a Different Subset of Features

In this study, we carried out a further comparative analysis to find out the best performing ensembles in Section 2. First, we compared the performance of various ensemble methods using sampling techniques (SMOTE and RUS) in Section 4.3. Moreover, we compared the performance of the ensemble with different sizes of ensembles in Section 4.4. Finally, we achieved the best-performing ensemble, AdaBoost, RF, and RUS Boost, to further experiment with feature selection in Section 4.4. 

Feature selection is a technique to reduce the redundant features and extract the discriminatory features. We used two popular feature selection methods, information gain and lasso, to select the optimal features in this experiment. The 10 best subsets of features obtained using lasso and information gain are presented in Table 10. We can observe from the table that lasso feature selection achieved baseline features, TQWT features, bandwidth parameters. By contrast, information gain feature selection achieved bandwidth parameters, formant frequencies, baseline features, and TQWT features, which are essential features to classify the PD. In addition, comparative performance evaluation with a different set of features using information gain was trained and tested on three different ensemble classifiers (AdaBoost, RF, and RUSBoost), which are shown in Figure 2a–c.

AdaBoost performed best with 10 features with the value 0.903 in F1-score using information gain feature selection methods, whereas random forest performed best with 16 features with 0.883 in AUROC. Similarly, RUSBoost performed best with six features with the value 0.79 in G-mean. Further, we performed feature selection with lasso. The comparative evaluation of performance measures is shown in Figure 3a–c. AdaBoost performed best with 16 features with the value 0.917 in the F1-score using lasso feature selection methods, whereas RF performed best with 15 features with the value 0.905 in AUROC. Similarly, RUSBoost performed best with ten features with the value 0.77 in G-Mean.

Finally, we conclude that AdaBoost performs best with 10 features using information gain. The comparative performance evaluation shows 10 features versus all features in Figure 4a,b. From Figure 4a, we can observe that IG with RF, IG with AdaBoost, and IG with RUSBoost in terms of F1-score measure ten features against all features have marginally differenced 0.039, 0.034, and 0.037, respectively.

Similarly, there exist marginal differences in terms of AUROC (0.072, 0.067 and 0.037) and G-Mean (0.07, 0.08 and 0.11). We can also observe from Figure 4a that lasso with RF, lasso with AdaBoost, and lasso with RUSBoost in terms of F1-score measure 10 features against all features to have marginally differenced 0.034, 0.041, and 0.050, respectively. Similarly, there exist marginal differences in terms of AUROC (0.077,0.070 and 0.056) and G-Mean (0.06, 0.10 and 0.06).

## 5. Conclusions and Future Scope

The present study is a comprehensive study using a machine learning approach, specifically that of the decision tree-based ensembles used in the imbalanced PD vocal-based speech dataset. A comparative study was done with oversampling and without it using the SMOTE technique in different ratios and different ensemble sizes. The performance metrics in this experiment are the F1-score, Precision, Recall, AUPRC, AUROC, and the G-Mean. The results of the model evaluation metrics suggest that appropriate performance metrics must be used to evaluate the performance of a classifier built on datasets with class imbalance problems; otherwise, results could be misinformative. Furthermore, the results show that the AdaBoost, randon forest, and RUSBoost ensemble methods performed best. Further, 10 best-performing features were selected in the present study with two feature selection methods: lasso and information gain. The baseline features, TQWT features, bandwidth parameters, and formant frequencies are important features to classify PD. Finally, the AdaBoost ensemble classifier performing best with the 10 features selected using the feature selection method of information gain. In the future, we will compare our present classifiers with some other types of ML classifiers, such as convolutional neural networks (CNN) and ensemble-based classifiers using Support Vector Machines (SVM). We will carry out a similar study with other datasets.

## Figures and Tables

**Figure 1 diagnostics-12-03000-f001:**
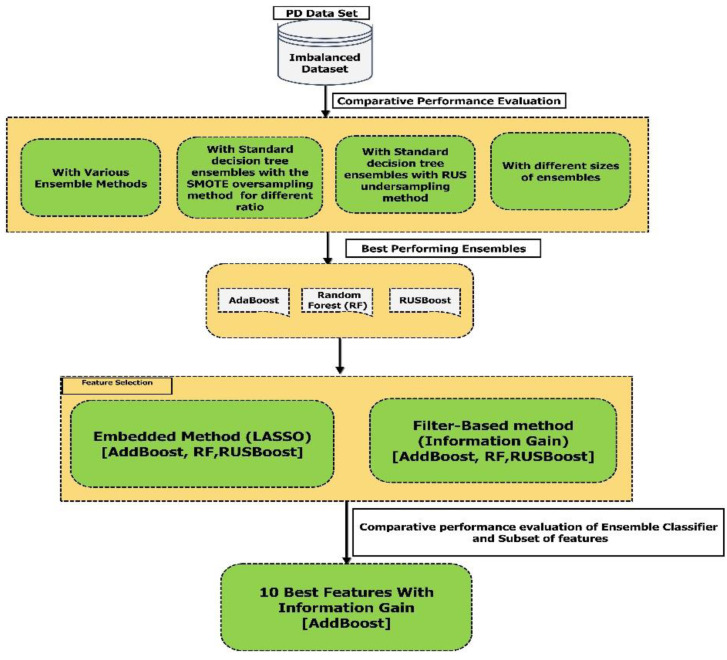
The methodology implemented in this research for comparative performance evaluation of ensemble classifiers (oversampling, undersampling), feature selection, model building.

**Figure 2 diagnostics-12-03000-f002:**
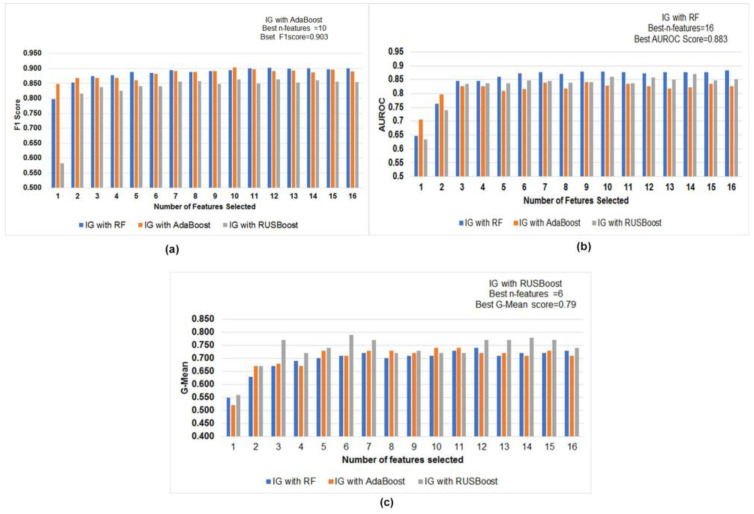
(**a**–**c**): Comparative evaluation of performance measures (**a**) F1-score (**b**) AUROC (**c**) G-mean in models built using a different subset of features using information gain.

**Figure 3 diagnostics-12-03000-f003:**
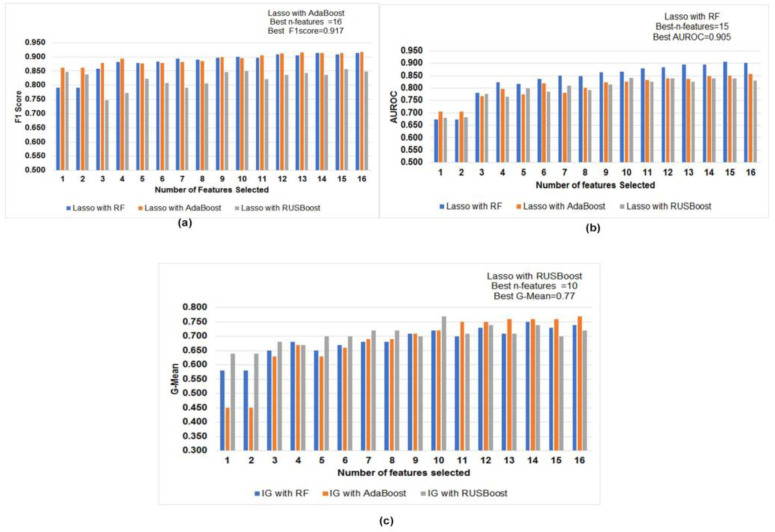
(**a**–**c**): Comparative evaluation of performance measures (**a**) F1-score (**b**) AUROC (**c**) G-Mean in models built using a different subset of features using lasso.

**Figure 4 diagnostics-12-03000-f004:**
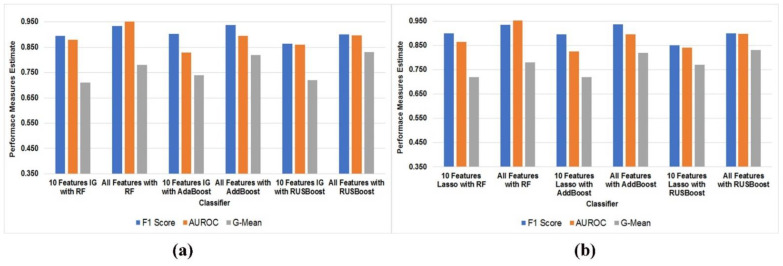
(**a**,**b**): Comparative evaluation of performance measures F1-score, AUROC, G-Mean in models built using 10 features against all features set using information gain and lasso.

**Table 1 diagnostics-12-03000-t001:** Parkinson’s speech dataset features.

Features	Measure	Number of Features
Baseline features	Jitter variants	5
	Shimmer variants	6
	Fundamental frequency parameters	5
	Harmonicity parameters	2
	Recurrence Period Density Entropy (RPDE)	1
	Detrended Fluctuation Analysis (DFA)	1
	Pitch Period Entropy (PPE)	1
Time-Frequency Features	Intensity Parameters	3
	Formant Frequencies	4
	Bandwidth	4
Tunable Q-factor Wavelet Transform (TQWT)	TQWT features related to F_0_	432
Wavelet Transform based Features	Wavelet Transform (WT) features related to F_0_	182
Vocal fold features	Glottis Quotient (GQ)	3
	Glottal to Noise	6
	Vocal Fold Excitation Ratio (VFER)	7
	Empirical Mode Decomposition (EMD)	6
Mel Frequency Cepstral Coefficients (MFCCs)	MFCCs	84

**Table 2 diagnostics-12-03000-t002:** Experimental setup using packages and classifier.

Classifier (Decision Tree Ensembles)	Software Package
Bagging	Weka tool
C4.5 Decision tree	Weka tool
AdaBoost	Weka tool
Random forest (RF)	Weka tool
Balanced random forset	Imblearn (Python)
XGBoost	Python
Balanced Bagging	Imblearn (Python)
RUSBoost	Isolation Forset
Isolation Forset	Weka tool
Random under sampling with bagging (RUSBagging)	Filter (SpreadsubSample) (Weka), Weka tool
Random under sampling with Random Forest (RUS random forest)	Filter (SpreadsubSample) (Weka), Weka tool
Random under sampling with XGBoost (RUS XGBoost)	Imblearn (Python), XGBoost
Random under sampling with AdaBoost (RUS AdaBoost)	Filter (SpreadsubSample) (Weka), Weka tool
Oversampling with Random Froest (SMOTE RF)	SMOTE, Weka Tool
Oversampling with Bagging (SMOTE Bagging)	SMOTE, Weka Tool SMOTE, Weka Tool
Oversampling with XGBoost (SMOTE XGBoost)	Imblearn (Python), XGBoost (Python)
Oversampling with AdaBoost (SMOTE AdaBoost)	SMOTE, Weka Tool

**Table 3 diagnostics-12-03000-t003:** The comparative study of different ensemble classifiers. Various types of performance metrics are used to measure ensembles performance. (Best results are shown in bold numbers).

Ensemble	Accuracy	AUROC	SN	SP
Single decision tree [J48]	0.855	0.848	0.896	0.730
Bagging	0.882	0.940	0.948	0.676
Random forest [RF]	0.895	**0.952**	**0.983**	0.622
XGBoost	0.800	0.928	0.972	0.625
AdaBoost	**0.901**	0.895	0.965	0.703
**Ensembles for imbalanced datasets**				
Balanced random forest	0.820	0.897	0.844	**0.792**
BalancedBagging	0.780	0.883	0.794	0.750
RUSBoost	0.830	0.897	0.979	**0.792**
Isolation forest	0.750	0.567	0.974	0.541

**Table 4 diagnostics-12-03000-t004:** The performance metric of ensembles (standard decision tree) along with oversampling method (SMOTE) of minority and majority class points for different ratios. SN, SP, and G-Mean performance metrics are considered for this experiment. (Best results are shown in bold numbers).

Ensemble	Original			Ratio = 1			Ratio = 0.75			Ratio = 0.50		
	SN	SP	G-Mean	SN	SP	G-Mean	SN	SP	G-Mean	SN	SP	G-Mean
Single decision tree	0.896	0.730	0.810	0.861	0.568	0.690	0.878	0.541	0.690	0.896	0.649	0.760
Bagging	0.948	0.675	0.800	0.930	0.784	0.850	0.922	0.703	0.800	0.913	0.676	0.790
Random forest	**0.983**	0.622	0.780	0.913	0.730	0.820	0.939	0.703	0.810	0.939	**0.784**	0.860
XGBoost	0.972	0.625	0.780	0.915	0.667	0.780	0.950	0.667	0.830	0.957	0.604	0.760
AdaBoost	0.965	0.703	0.82	0.948	0.703	0.820	0.957	**0.784**	**0.870**	0.948	0.757	0.850

**Table 5 diagnostics-12-03000-t005:** The performance metric of ensembles (standard decision tree) along with oversampling method (SMOTE) of minority and majority class points for different ratios. AUROC and AUPRC performance metrics are considered for this experiment. (Best results are shown in bold numbers.).

Ensemble	Original		Ratio = 1		Ratio = 0.75		Ratio = 0.50	
	AUROC	AUPRC	AUROC	AUPRC	AUROC	AUPRC	AUROC	AUPRC
Single decision tree	0.848	0.914	0.752	0.871	0.727	0.848	0.760	0.875
Bagging	0.940	0.978	0.953	0.986	0.943	0.983	0.946	0.984
Random forest	0.952	0.984	0.949	0.982	0.953	0.984	**0.963**	**0.988**
XGBoost	0.928	0.974	0.940	0.980	0.930	0.976	0.927	0.974
AdaBoost	0.895	0.941	0.951	0.981	0.940	0.967	0.925	0.959

**Table 6 diagnostics-12-03000-t006:** The performance metric of ensembles (standard decision tree) along with the undersampling method (RUS) of minority and majority class points for different ratios. (Bold numbers show the best results.) AUROC and AUPRC performance metrics are considered for this experiment.

Ensemble	Original		Ratio = 1		Ratio = 0.75		Ratio = 0.50	
	AUROC	AUPRC	AUROC	AUPRC	AUROC	AUPRC	AUROC	AUPRC
Single decision tree	0.848	0.914	0.780	0.888	0.869	0.938	0.772	0.884
Bagging	0.901	0.960	0.944	0.983	0.945	0.983	0.919	0.974
Random forest	0.967	0.990	0.928	0.977	0.943	0.981	0.945	0.981
XGBoost	0.928	0.974	0.893	0.963	0.907	0.969	0.917	0.972
AdaBoost	0.937	0.981	0.947	0.983	**0.954**	**0.986**	0.922	0.962

**Table 7 diagnostics-12-03000-t007:** The performance metric of ensembles (standard decision tree) along with the undersampling method of minority and majority class points for different ratios SN, SP, and G-Mean performance metrics are considered for this experiment. (Bold numbers show the best results.).

Ensemble	Original			Ratio = 1			Ratio = 0.75			Ratio = 0.50		
	SN	SP	G-Mean	SN	SP	G-Mean	SN	SP	G-Mean	SN	SP	G-Mean
Single decision tree	0.896	**0.730**	0.810	0.739	0.811	0.770	0.809	0.784	0.800	0.861	0.622	0.730
Bagging	0.948	0.676	0.800	0.844	**0.892**	**0.870**	0.878	0.838	0.800	0.887	0.703	0.790
Random forest	**0.983**	0.622	0.780	0.844	0.811	0.830	0.870	0.784	0.830	**0.939**	0.622	0.760
XGBoost	0.972	0.625	0.780	0.804	0.744	0.770	0.902	0.769	0.830	0.920	0.744	0.830
AdaBoost	0.965	0.703	**0.820**	0.852	0.865	0.860	0.878	0.838	0.860	0.922	0.784	0.850

**Table 8 diagnostics-12-03000-t008:** The AUROC performance metric of different ensembles of classifiers (decision trees) with various sizes of ensembles (Bold numbers show the best results.).

Ensemble	20	50	100	200
Bagging	**0.941**	0.940	0.941	0.941
Random forest	0.940	**0.952**	**0.967**	**0.969**
AdaBoost	0.940	0.895	0.850	0.857
XGBoost	0.919	0.928	0.929	0.927
Balanced random forest	0.872	0.897	0.913	0.910
BalancedBagging	0.892	0.883	0.863	0.872
RUSBoost	0.916	0.922	0.932	0.938
Isolation forest	0.566	0.567	0.571	0.550

**Table 9 diagnostics-12-03000-t009:** The AUPRC, SN, SP, G-Mean performance metrics of different ensembles of classifiers (decision trees) with various sizes of ensembles (Bold numbers show the best results.).

Ensemble		20				50				100				200		
	AUPRC	SN	SP	G-Mean	AUPRC	SN	SP	G-Mean	AUPRC	SN	SP	G-Mean	AUPRC	SN	SP	G-Mean
Bagging	**0.980**	0.965	0.703	0.820	0.978	0.948	0.676	0.800	0.979	0.948	0.649	0.780	0.980	0.957	0.595	0.750
Random forest	0.979	0.966	0.622	0.770	**0.984**	**0.983**	0.622	0.780	**0.990**	0.974	0.595	0.760	**0.990**	0.965	0.541	0.770
AdaBoost	0.976	0.965	0.649	0.790	0.941	0.965	0.703	0.820	0.916	0.974	0.649	0.790	0.917	**0.983**	0.703	0.830
XGBoost	0.970	0.955	0.615	0.770	0.974	0.972	0.625	0.780	0.974	0.884	0.718	0.800	0.974	0.893	0.769	0.830
Balanced random forest	0.955	0.773	0.750	0.760	0.963	0.844	0.792	0.820	0.970	0.830	**0.813**	0.820	0.969	0.837	0.792	0.810
BalancedBagging	0.957	0.837	0.770	0.800	0.959	0.794	0.750	0.790	0.947	0.844	0.771	0.810	0.956	0.851	0.750	0.800
RUSBoost	0.966	0.872	0.770	0.820	0.970	0.979	0.792	0.830	0.973	0.943	0.771	0.850	0.976	0.972	0.771	**0.870**
Isolation forest	0.819	0.948	0.541	0.230	0.807	0.974	0.541	0.230	0.830	0.965	0.541	0.230	0.815	0.974	0.541	0.230

**Table 10 diagnostics-12-03000-t010:** The 10 subsets of features using information gain and lasso with rank and coefficient.

Rank	10 Best Feature Selected with Information Gain	Coefficient	10 Best Feature Set with Lasso	Common Feature in Both Feature Selection
0.1398	std_6th_delta_delta	3.181598	std_6th_delta_delta	std_6th_delta_delta
0.139	std_delta_delta_log_energy	2.406294 × 10^−1^	std_delta_delta_log_energy	std_delta_delta_log_energy
0.1371	mean_MFCC_2nd_coef	2.559498 × 10^−2^	mean_MFCC_2nd_coef	mean_MFCC_2nd_coef
0.1324	std_delta_log_energy	3.515898 × 10^−7^	tqwt_entropy_log_dec_26	
0.1321	tqwt_TKEO_mean_dec_12	2.720495 × 10^−1^	tqwt_minValue_dec_12	
0.1311	tqwt_entropy_log_dec_11	2.045678	std_7th_delta_delta	
0.1282	tqwt_entropy_shannon_dec_11	3.281899	std_9th_delta_delta	
0.1258	tqwt_stdValue_dec_11	−1.039598	tqwt_stdValue_dec_11	tqwt_stdValue_dec_11
0.1239	std_8th_delta_delta	−3.670655 × 10^−4^	tqwt_kurtosisValue_dec_27	
0.1233	tqwt_entropy_shannon_dec_12	−2.894499 × 10^−4^	tqwt_kurtosisValue_dec_26	

## Data Availability

Data is publicly avaible.

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
