# Peer review of "Analysis of Parkinson’s Disease Using an Imbalanced-Speech Dataset by Employing Decision Tree Ensemble Methods"

_diagnostics, 2022, doi:10.3390/diagnostics12123000_

Round 1

Reviewer 1 Report

In the study, the authors study the vocal impairment feature dataset of PD, which is imbalanced. The authors propose the comparative performance evaluation using various decision tree ensembles methods with or without oversampling technique. In addition, the authors compare the performance of classifiers with different sizes of ensembles and various ratios of minority class and majority class with oversampling and undersampling. Finally, the authors combine feature selection with best-performing ensembles classifiers. The result shows that AdaBoost, Random Forest, and decision tree developed for the imbalanced dataset, i.e., RUSBoost, perform well in performance metrics such as precision, recall, F1 measure, AUROC and geometric mean.

The paper is interesting but lacks scientific novelty. Moreover, the motivation of the paper is not clear. The authors miss a lot of technical details. For example, the data preprocessing stage is not well-described. Why don't the authors present time-frequency analysis of the data?

The titles of the paper's subparts are confusing (for example, 4.1 and 4.2). The paper should be strongly edited. 

The paper should be proofread. There are a lot of typos in the paper.

Author Response

Original Manuscript ID: diagnostics-1788208      

Original Article Title: “Detecting Earlier Stages of Parkinson Disease using Imbalanced-Speech Dataset by Employing Decision Tree Ensemble Methods.”

To: Diagnostics Editor

Re: Response to reviewers

Dear Editor,

Thank you for providing us the opportunity to submit a response letter along with the updated manuscript which we have carefully prepared after reading the reviewers comments. We would like to thank the reviewers for their constructive feedback and valuable comments which have resulted in improvements to the manuscript.

We are uploading (a) our point-by-point response to the comments (below) (response to reviewers).

Best regards,

Omar Barukab et al.

Reviewer#1, Concern # 1: The paper is interesting but lacks scientific novelty

Author response:  We have added following lines in the paper (on page 2)

” Not many researchers (36,56-59) analyze the data by considering the data as imbalance data. In this paper we have done extensive study on this dataset using the algorithms developed for imbalance data.

 Following are the contributions of our work;

1- We have considered the data as imbalanced data and carried out an extensive study.

2- Decision tree ensembles have shown excellent performances in different domains. In this study, we carried out extensive performance evaluations of different types of decision tree ensembles including those that are developed for imbalanced data.”

Reviewer#1, Concern # 2: the motivation of the paper is not clear.

Author response:  We discussed the motivation of the paper in related work sub section (pages 10-11)

Reviewer#1, Concern # 3: The authors miss a lot of technical details. For example, the data preprocessing stage is not well-described.

Author response:  We discussed the data preprocessing part in section 4.1 (on page 15).

“This experiment oversampled the minority class, and we used SMOTE technique using Weka. Similarly, for Random Undersampling, we used SpreadsubSample supervised filter (Weka) and Imblearn (python) package. We implemented RUS undersampling using weka and imblearn package with the majority class to change the ratio of majority class to majority class.”

Reviewer#1, Concern # 4: Why don't the authors present time-frequency analysis of the data.

Author response:   Thanks for the suggestion. We added the following text in the Related work Section (pages 2-3)

“There are different approaches to analyze this data set, deep learning, machine learning, time-series analysis etc.), however, in this paper, we concentrate only on decision tree ensembles.”

Reviewer#1, Concern # 5: The titles of the paper's subparts are confusing (for example, 4.1 and 4.2). The paper should be strongly edited.

Author response:  The concern # 5 of the reviewer has been executed. Thanks for the suggestion. We removed all the typos error in manuscript and change the paper’s subpart. 4.1 (on page 14) and 4.2 (on page 15)

Reviewer 2 Report

The manuscript is interesting. As suggestions for improvement, I recommend reducing the size, it is difficult to read, I recommend combining tables. Table 11 I suggest elaborating it in a simpler way.

It is an interesting article, which I suggest is simpler or more synthetic.

Author Response

Original Manuscript ID: diagnostics-1788208      

Original Article Title: “Detecting Earlier Stages of Parkinson Disease using Imbalanced-Speech Dataset by Employing Decision Tree Ensemble Methods.”

To: Diagnostics Editor

Re: Response to reviewers

Dear Editor,

Thank you for providing us the opportunity to submit a response letter along with the updated manuscript which we have carefully prepared after reading the reviewers comments. We would like to thank the reviewers for their constructive feedback and valuable comments which have resulted in improvements to the manuscript.

We are uploading (a) our point-by-point response to the comments (below) (response to reviewers).

Best regards,

Omar Barukab et al.

Reviewer#2, Concern # 1: The manuscript is interesting. As suggestions for improvement, I recommend reducing the size, it is difficult to read, I recommend combining tables. Table 11 I suggest elaborating it in a simpler way.

Author response:  Thanks for the suggestion we combine table 9 into 10. In addition, we reduce the size of introduction section (pages 1 - 4) and related work section (pages 5 - 9).

Reviewer 3 Report

The paper uses an unbalanced dataset to address the automatic classification of Parkinson's disease (PD) from acoustic signals.

In my opinion, the research topic is of great interest to the community, as many works do not address the problem of unbalanced datasets and the influence of the machine learning methods used for classification. However, I found the paper very difficult to follow, the information presented is not clear, and the wording and grammar should be carefully revised.

Major concerns:

-       There is no clarity in the contributions of the study. For instance, the first contributions states that:

“We have considered the data as imbalanced data and carried out an extensive study".

o   First of all, you have to specify what kind of study was carried out, so the reader knows what is the contribution of you work.

o   Second, you did not “considered the data as imbalanced data”, you “considered a unbalanced dataset”. The wording is used in the original sentence suggest that you assumed something form the dataset, which is just something very strange to do.

o   Considering the previous comments, I would rather say something like “We considered an unbalanced dataset and performed automatic classification between PD patients and healthy controls to evaluate the robustness of different ensemble methods for class unbalance”.

-       I’m also not sure exactly what is the contribution of you work. In section 2. Related Work, there are plenty of studies that also used unbalanced datasets. Thus, what is the novelty of your work and how is it different from the works reported in the state-of-the-art?

Lines 72 to 75: I do not understand what you are trying to say. How come the detection of PD improves when non-invasive techniques are used? This is a very odd argument. Is there a source for this? In the same paragraph is mentioned that in the current study a dataset with 188 patients and 64 healthy controls was considered. How is this related to the previous claim of PD assessment being improved when non-invasive methods are used?

-       In line 104 the authors say that only ensemble methods are used, but never explained why only this. Please support this with an argument.

-       In lines 132-134 is mentioned that only works that have used the same dataset as you, are considered for review. However, in lines 144-150 two completely datasets are mentioned; thus, the results are not comparable with the other studies. So, what exactly is the idea in this section?

-       Line 183: “Auto encoders are not commonly used as feature extractors”. Well I think this is not correct. In many applications, autoencoders are trained with the purpose of computing bottleneck features when test data is used.

-       Lines 238-239: “The dataset used in earlier studies has a problem of class imbalance because out of 195 cases, 147 are connected to PD and the rest are related to healthy people”. This is the beginning of a paragraph, and I don’t understand how is this connected to the previous text nor the one following. Also what study?, In lines132-134 was mentioned that only the Istanbul dataset was going to be considered. Please clarify.

-       Are the oversampling and undersampling approaches a form of data augmentation? If so, you should just mention it. Data augmentation is a common term used in machine learning when there is no enough data available to train a classifier/regressor.

-       The oversampling and undersampling approaches are addressed in section 3.4, but is not clear how the data is generated artificially. Is it by adding noise to the recordings? changing pitch? other? Please clarify.

-       A lot of methods are mentioned, but no explained. For instance, “artificial bee colony-based features” (line 172); “binary Rao” (line208), IGKNN; MCC…Many of these methods are not known, thus, a brief explanation should be sufficient.

-       In lines 292-293 it is mentioned that the gender is included in the feature vector used for automatic classification. This is a serious methodological error, as you are including information about the class of the subject, thus, biasing the classifier towards a target value.

-       I don’t understand from where the numbers in lines 535, 539, and 543 come from. What does the number mean?

Minor concerns:

-       Please be consistent with the way the information is presented. Usually, the accuracy, specificity, and sensitivity are presented in %. In the paper, sometimes those measurements are presented in % and sometime are presented as decimal points.

-       Al so be consistent with the name of the “F1-score”. In the paper is called F-1, F1Score, F1.score, F1-Score, F1-measure.

-       Line 176: comparisioned->comparison

-       Line 157-162: Please revised this whole paragraph.

-       Line 247: Elated->Related

-       Line 288: “a microphone’s frequency response was set to 44.1kHz”. I think you meant to say that the recordings were sampled a 44.1kHz.

-       Line 290: “the vowel “a” letter was collected…”. Please rewrite this.

-       Abbreviations are defined only once, however, Parkinson’s disease (PD) is define multiple times in the whole document. Same for sensitivity, specificity, decision trees, and others.

-       Tables 4-8: “matric” -> metric.

Author Response

Reviewer#3, Concern # 1:

Major concerns:

There is no clarity in the contributions of the study. For instance, the first contributions states that: “We have considered the data as imbalanced data and carried out an extensive study".

  • First of all, you have to specify what kind of study was carried out, so the reader knows what is the contribution of you work.
  • Second, you did not “consider the data as imbalanced data”, you “considered a unbalanced dataset”. The wording is used in the original sentence suggest that you assumed something form the dataset, which is just something very strange to do.
  • Considering the previous comments, I would rather say something like “We considered an unbalanced dataset and performed automatic classification between PD patients and healthy controls to evaluate the robustness of different ensemble methods for class unbalance”.

Author response: Contributions have been rewritten (Highlighted by yellow color in the manuscript), and they are as follows:

  1. We considered an Imbalanced dataset and performed automatic classification between PD patients and healthy controls to evaluate the robustness of different ensemble methods for class Imbalance.
  2. Decision tree ensembles have been shown to have excellent performance in different domains. In this study, we carried out extensive performance evaluations of different types of decision tree ensembles such as RUSBoost, Isolation Forest, RUS Bagging, Balanced Bagging, etc. Those that are developed for imbalanced data. To the best of our knowledge, which has never been used by other researchers in this area.
  3. We carried out the feature selection using Lasso and Information gain method to achieve the best set of features.

Response Regarding the second comment

In machine learning, whenever there is a large ratio between majority class and minority class generally, we call it an imbalanced dataset. Majority of paper used imbalanced as title [68][69][71][72][74][75][83][84][86][87].

Reviewer#3, Concern # 2:

I’m also not sure exactly what is the contribution of you work. In section 2. Related Work, there are plenty of studies that also used unbalanced datasets. Thus, what is the novelty of your work and how is it different from the works reported in the state-of-the-art?

Author response:

Our Work concentrated on the PD speech signal dataset [60]. Some people have worked on this dataset and generally, they have used SMOTE method however we have used along with SMOTE many other methods such as RUSBoost, Isolation Forest, RUS Bagging, Balanced Bagging etc. Those that are developed for imbalanced dataset. This is more detail study in this area.

Reviewer#3, Concern # 3:

Lines 72 to 75: I do not understand what you are trying to say. How come the detection of PD improves when non-invasive techniques are used? This is a very odd argument. Is there a source for this? In the same paragraph is mentioned that in the current study a dataset with 188 patients and 64 healthy controls was considered. How is this related to the previous claim of PD assessment being improved when non-invasive methods are used?

Author response:  Literature review has been rewritten (Highlighted by yellow color in the manuscript). Thanks for the suggestion.

Reviewer#3, Concern # 4:

In line 104 the authors say that only ensemble methods are used, but never explained why only this. Please support this with an argument.

Author response: 

Ensemble methods have shown very good results and very robust result. As per the literature ensemble methods were used to explore the imbalanced dataset in various domain. However, these ensemble methods have not been applied to PD dataset. Therefore, we intend to extensively explore the decision tree ensembles such as balanced random forest (RUS), Synthetic Minority Oversampling Technique (SMOTE) Bagging, RUS bagging, SMOTE Boost, RUSBoost, and so forth to predict the PD in its earlier stages.

Reviewer#3, Concern # 5:

In lines 132-134 is mentioned that only works that have used the same dataset as you, are considered for review. However, in lines 144-150 two completely datasets are mentioned; thus, the results are not comparable with the other studies. So, what exactly is the idea in this section?

Author response:  Literature review has been rewritten (Highlighted by yellow color in the manuscript). Thanks for the suggestion.

Reviewer#3, Concern # 6:

Line 183: “Auto encoders are not commonly used as feature extractors”. Well I think this is not correct. In many applications, autoencoders are trained with the purpose of computing bottleneck features when test data is used.

Author response: The literature review has been rewritten (Highlighted by yellow color in the manuscript). Thanks for the suggestion.

Reviewer#3, Concern # 7:

Lines 238-239: “The dataset used in earlier studies has a problem of class imbalance because out of 195 cases, 147 are connected to PD and the rest are related to healthy people”. This is the beginning of a paragraph, and I don’t understand how is this connected to the previous text nor the one following. Also what study?, In lines132-134 was mentioned that only the Istanbul dataset was going to be considered. Please clarify.

Author response: The literature review has been rewritten (Highlighted by yellow color in the manuscript). Thanks for the suggestion.

Reviewer#3, Concern # 8:

Are the oversampling and undersampling approaches a form of data augmentation? If so, you should just mention it. Data augmentation is a common term used in machine learning when there is insufficient data available to train a classifier/regressor.

Author response: 

Oversampling increases the size of training data, whereas undersampling decreases the size of data. Oversampling is data augmentation However, undersampling is not data augmentation. Data augmentation means increasing the size of data.

Reviewer#3, Concern # 9:

The oversampling and undersampling approaches are addressed in section 3.4 but is not clear how the data is generated artificially. Is it by adding noise to the recordings? changing pitch? other? Please clarify.

Author response:  In oversampling data generated through SMOTE technique. SMOTE generates synthetic data by utilizing a k-nearest neighbor algorithm. SMOTE selects random data from the minority class, then select k-nearest neighbors from the data. Artificially data would then be generated between the random data and the randomly selected k-nearest neighbor.

Reviewer#3, Concern # 10:

A lot of methods are mentioned, but no explained. For instance, “artificial bee colony-based features” (line 172); “binary Rao” (line208), IGKNN; MCC…Many of these methods are not known, thus, a brief explanation should be sufficient.

Author response: The literature review has been rewritten (Highlighted by yellow color in the manuscript). Thanks for the suggestion.

Reviewer#3, Concern # 11:

In lines 292-293 it is mentioned that the gender is included in the feature vector used for automatic classification. This is a serious methodological error, as you are including information about the class of the subject, thus, biasing the classifier towards a target value.

Author response: 

This dataset has many features, and we have used the same features as suggested by the dataset’s author. This is one of the features it is not that we have decided it. Basically, this was a feature in the dataset given by the author of the dataset.

Reviewer#3, Concern # 12:

I don’t understand from where the numbers in lines 535, 539, and 543 come from. What does the number mean?

Author response: 

These line numbers are the marginal difference between 10 features vs. All features of each classifier

For example

All features with RF (0.934) -IG with RF (0.895) =Marginal difference (0.039), Similarly rest values are presented in table

IG with RF

All Features with RF

IG with AdaBoost

All Features with AddBoost

IG with RUSBoost

All Features with RUSBoost

F1 Score

0.895

0.934

0.903

0.937

0.863

0.900

AUROC

0.88

0.952

0.828

0.895

0.860

0.897

G-Mean

0.710

0.78

0.740

0.82

0.720

0.83

Marginal Differences

0.934-0.895

 =0.039

0.937-0.903=

0.034

0.900-.863=

0.037

0.952-0.88

 =0.072

0.895-.828=

0.067

0.897-.860=

0.037

0.78-0.710

 =0.070

0.82-0.740=

0.080

0.83-0.720=

0.110

Similarly

Lasso with RF

All Features with RF

Lasso with AddBoost

All Features with AddBoost

Lasso with RUSBoost

All Features with RUSBoost

F1 Score

0.900

0.934

0.896

0.937

0.850

0.900

AUROC

0.865

0.952

0.825

0.895

0.841

0.897

G-Mean

0.720

0.78

0.720

0.82

0.770

0.83

Marginal Difference

0.034

0.041

0.050

0.087

0.070

0.056

0.060

0.100

0.060

Reviewer#3, Concern # 13:

Minor concerns:

Please be consistent with the way the information is presented. Usually, the accuracy, specificity, and sensitivity are presented in %. In the paper, sometimes those measurements are presented in % and sometime are presented as decimal points.

Author response: Concern # 13 of the Reviewer has been executed. Thanks for the suggestion.

Reviewer#3, Concern # 14:

Al so be consistent with the name of the “F1-score”. In the paper is called F-1, F1Score, F1.score, F1-Score, F1-measure.

Author response: Concern # 21 of the Reviewer has been executed. Thanks for the suggestion.

Reviewer#3, Concern # 15:

Line 176: comparisioned->comparison

Author response: Concern # 21 of the Reviewer has been executed. Thanks for the suggestion.

Reviewer#3, Concern # 16:

Line 157-162: Please revise this whole paragraph.

Author response: Concern # 13 of the Reviewer has been executed (Highlighted by yellow color in the manuscript). Thanks for the suggestion.

Reviewer#3, Concern # 17:

Line 247: Elated->Related

Author response:  The concern # 21 of the Reviewer has been executed. Thanks for the suggestion.

Reviewer#3, Concern # 18:

Line 288: “a microphone’s frequency response was set to 44.1kHz”. I think you meant to say that the recordings were sampled a 44.1kHz.

Author response:  yes

Reviewer#3, Concern # 19:

Line 290: “the vowel “a” letter was collected…”. Please rewrite this.

Author response:  The concern # 21 of the Reviewer has been executed (Highlighted by yellow color in the manuscript). Thanks for the suggestion.

Furthermore, the vowel letter "a" pronounced by the PD patient was collected in replicates of three

Reviewer#3, Concern # 20:

Abbreviations are defined only once, however, Parkinson’s disease (PD) is define multiple times in the whole document. Same for sensitivity, specificity, decision trees, and others.

Author response:  The concern # 20 of the Reviewer has been executed. Thanks for the suggestion.

Reviewer#3, Concern # 21:

Tables 4-8: “matric” -> metric.

Author response:  The concern # 21 of the Reviewer has been executed. Thanks for the suggestion.

Reviewer 4 Report

In this work, the authors propose a decision tree ensemble method to help diagnose the Parkinson’s disease. The work has real potential due to this important application.

1. The paper is really hard to read. For example, there is no need to report the numbers such as the accuracy and the F1-score in Section 2.  It is hard to get any insight into these numbers without knowing what the data look like. It is also awkward to talk about many criteria when their introductions are given in later sections. (Some criterion like MCC is not even introduced in the later sections.) These numbers can be put into the numerical studies in later sections and only introduce the related work in this section.

2. Throughout the data, is the accuracy or other criteria like the F1-score calculated on only the training data or evaluated on a hold-out validation dataset? 

3. How is the tuning parameter handled in this work? Is any out-of-bag error used to help build random forest classifiers?

4. What is the advantage of using information gain to rank the important features, rather than using the variance important that has been implemented in boosting and random forest?

5. In Section 3.5.2, how is the lasso method used? Is the L1 penalty applied to, say, logistic regression?

Author Response

Reviewer#4, Concern # 1: 1. The paper is really hard to read. For example, there is no need to report the numbers such as the accuracy and the F1-score in Section 2.  It is hard to get any insight into these numbers without knowing what the data look like. It is also awkward to talk about many criteria when their introductions are given in later sections. (Some criterions like MCC is not even introduced in the later sections.) These numbers can be put into the numerical studies in later sections and only introduce the related work in this section.

Author response:  We present the same in our study therefore, in section 2, we mentioned the previous work with accuracy and F1 score. (Pages 4-6)

The concern regarding the MCC evaluation matrix, we removed from the related work section (pages 4-6)

Reviewer#4, Concern # 2: Throughout the data, is the accuracy or other criteria like the F1-score calculated on only the training data or evaluated on a hold-out validation dataset?

Author response:  We used test data to evaluate our model, which is mentioned in section 4.1 (on page 10).

Reviewer#4, Concern # 3: How is the tuning parameter handled in this work? Is any out-of-bag error used to help build random forest classifiers?

Author response:  We used default parameter as we mentioned in section 4.1 (on page 10).

Reviewer#4, Concern # 4: What is the advantage of using information gain to rank the important features, rather than using the variance important that has been implemented in boosting and random forest?

Author response:   There are many methods to rank the features. We achieved better result in Information gain. We are not claiming Information gain is best methods to rank the features.

Reviewer#4, Concern # 5: In Section 3.5.2, how is the lasso method used? Is the L1 penalty applied to, say, logistic regression?

Author response:  In the current study, we focus on the Least Absolute Shrinkage and Selection Operator (LASSO) or L1 regularization, an important example of an embedded technique to select the best subset of features significantly associated with the response variable. The LASSO approach shrinks the explanatory variables’ coefficients with less or no discriminatory power to zero while selecting a subset of explanatory variables with non-zero coefficients.

The LASSO is a clear case of the penalized least squares regression with lambda (λ) as an L1-penalty function. The tuning of the hyperparameter (penalty factor lambda) was performed during the cross-validation process. We applied the LASSO regression with 10-fold cross-validation (CV) to select the optimal subset of discriminatory attributes.

Round 2

Reviewer 1 Report

The authors try to address my comments but the paper still needs impovements. The authors' contribution is not enough to be accepted. My comments about technical details and time-frequency analysis are not addressed.

Author Response

Reviewer#1 (round 2), Concern # 1: The authors try to address my comments but the paper still needs improvements.

Author response:  Thanks for the encouraging comments of the reviewer regarding the improvement of the paper.

Reviewer#1 (round 2), Concern # 2: The authors' contribution is not enough to be accepted.

Author response:  We did a detailed study considering the data as imbalanced data which was not done before. We wrote our contributions in page 3. (Highlighted by yellow color in the manuscript)

  1. We considered an Imbalanced dataset and performed automatic classification between PD patients and healthy controls to evaluate the robustness of different ensemble methods for class Imbalance.
  2. Decision tree ensembles have been shown to have excellent performance in different domains. In this study, we carried out extensive performance evaluations of different types of decision tree ensembles such as RUSBoost, Isolation Forset, RUSBagging, Balanced Bagging, etc. Those that are developed for imbalanced data. To the best of our knowledge, which has never been used by other researchers in this area.
  3. We carried out the feature selection using Lasso and Information gain method to achieve the best set of features.

Reviewer#1 (round 2), Concern # 3: My comments about technical details and time-frequency analysis are not addressed.

Author response:  Technical details-

We discussed the required technical details herewith:

1-           We discussed the dataset in detail in section 3.1

2-           Classifiers are discussed in Sections 3.2-3.4.

3-           The feature selection method is discussed in Section 3.5

4-           Evaluation metrics are discussed in Section 3.6

5-           Section 4.1 has discussion software packages and experimental set-up.

6-           Table 2 presents software packages and related classifiers.

Reviewer#1 (round 2), Concern # 4: Time-frequency analysis

Author response:  In this paper, we did the study of decision tree-based machine learning models. The dataset used in the paper has different types of features including Time-Frequency features (Table 1). 

Reviewer 3 Report

I thank the authors for taking the time to read and address many of my suggestions; however, I still have some additional comments:

Comment #1:

In my first review I wrote (authors response: Reviewer#3, Concern#11): “In lines 292-293 it is mentioned that the gender is included in the feature vector used for automatic classification. This is a serious methodological error, as you are including information about the class of the subject, thus, biasing the classifier towards a target value.”

The author’s response was: “This dataset has many features, and we have used the same features as suggested by the dataset’s author. This is one of the features it is not that we have decided it. Basically, this was a feature in the dataset given by the author of the dataset.”

I understand that you want to compare with baseline features not defined by you; however, it does not mean that you should carry the same methodological error when doing your experiments. For example, using the “gender” information as a feature for a machine learning algorithm is a mistake because rather than an objective measurement, the gender is given by a human; thus, the system will be biased towards a “class label.” My suggestion, in this case, is NOT TO INCLUDE gender as a feature in the baseline, regardless if you did not suggest this.

Why am I so persistent with this? Because one of the conclusions (Section 5) stated that the baseline features are “important features to classify Parkinson’s disease”.

Comment #2:

Regarding the oversampling of data using SMOTE. I appreciated the explanation, but it is still not clear how the new data is generated. The authors wrote, “SMOTE selects random data from the minority class, then select k-nearest neighbors from the data. Artificially data would then be generated between the random data and the randomly selected k-nearest neighbor.”

What I understand from this is that you use kNN using the feature vectors from the underrepresented class (healthy controls) and then average these feature vectors with the ones that are close. If this is the case, then:

-      how many neighbors are considered?

-      how do you handle the “gender” feature?

Please clarify in the manuscript. To me, this is very interesting and, therefore, a technique that I would like to try myself.

Comment #3:

I noticed that in Table 3, the recall and the sensitivity (SN) are reported at the time. These two metrics are the same; thus, only one of them should be reported.

My main concern here is that the reported recall and SN are different for some ensemble methods:

-       XGBoost: Recall: 0.972; SN: 0.970

-       Balanced random forest: Recall: 0.840; SN: 0.844

-       BalancedBagging: Recall: 0.800; SN: 0.794

-       RUSBoost: Recall: 0.870; SN: 0.979

I agree that using different performance metrics is essential to have a better picture of the system; however, the tables are so dense, and the discussion in section 4 is so confusing that I suggest using only Accuracy, Sensitivity, Specificity, and AUC (ROC) to analyze and report your results. Given that you only have 2 classes, these four metrics are enough.

Also, do not limit the discussion to the AUC metric. The AUC is a good indicator of the separability of the two classes (PD vs. HC), but it does not tell how well the system classifies the samples. For instance, in Table 3, the AUC for Random Forest is 0.952, the Sensitivity 0.983, and the Specificity 0.622. This means that the classifier is very confident about the correctly classified samples (high AUC), but it diagnoses many of the HC as PD patients (low Specificity).

Minor comments:

The paper still needs some careful reading. The wording and grammar used in some sections make the manuscript very hard to read. Here are some examples, but I cannot do it for the whole document because it is simply too much work.

-       The abbreviation PD should also be defined in line 34. Additionally, the use of “PD” is very random in the current version of the document.

-       Line 99: Imbalanced -> imbalanced

-       Line 425: while->While      

-       Line 281-282: “Furthermore, the vowel letter "a" pronounced by the PD patient was collected in replicates of three” -> All participants were asked to perform the sustained phonation of the vowel “ah” three times.

-       Line 282: Speech effect is -> Speech disorders are.

-       Figure 1 is too small. Zooming-in does not help because the figure has a low resolution.

Author Response

Comment #1:

In my first review I wrote (authors response: Reviewer#3, Concern#11): “In lines 292-293 it is mentioned that the gender is included in the feature vector used for automatic classification. This is a serious methodological error, as you are including information about the class of the subject, thus, biasing the classifier towards a target value.”

The author’s response was: “This dataset has many features, and we have used the same features as suggested by the dataset’s author. This is one of the features it is not that we have decided it. Basically, this was a feature in the dataset given by the author of the dataset.”

I understand that you want to compare with baseline features not defined by you; however, it does not mean that you should carry the same methodological error when doing your experiments. For example, using the “gender” information as a feature for a machine learning algorithm is a mistake because rather than an objective measurement, the gender is given by a human; thus, the system will be biased towards a “class label.” My suggestion, in this case, is NOT TO INCLUDE gender as a feature in the baseline, regardless if you did not suggest this.

Why am I so persistent with this? Because one of the conclusions (Section 5) stated that the baseline features are “important features to classify Parkinson’s disease”.

Author response #1: 

According to datasets baseline features includes following features as mentioned in Table 1. The feature gender is an independent feature and gender does not come under the category of baseline features.

Features

Measure

Number of features

Baseline features

Jitter variants

5

Shimmer variants

6

Fundamental frequency parameters

5

Harmonicity parameters

2

Recurrence Period Density Entropy (RPDE)

1

Detrended Fluctuation Analysis (DFA)

1

Pitch Period Entropy (PPE)

1

In conclusion section we have stated that “baseline features are important features to classify Parkinson disease” because the top 10 features obtained using lasso and information gain comprises of feature subsets from the baseline category of features as mentioned in Table 10 and the baseline features obtained in Table 10 do not include the gender feature. Therefore, gender is not part of the final list of selected features used for building model to classify PD. (As mentioned in Table 10) (Page 14) (Highlighted by yellow color in the manuscript). Classification results with 10 features presented in figure 4 (a) and 4(b) do not use gender feature.

Comment #2:

Regarding the oversampling of data using SMOTE. I appreciated the explanation, but it is still not clear how the new data is generated. The authors wrote, “SMOTE selects random data from the minority class, then select k-nearest neighbors from the data. Artificially data would then be generated between the random data and the randomly selected k-nearest neighbor.”

What I understand from this is that you use kNN using the feature vectors from the underrepresented class (healthy controls) and then average these feature vectors with the ones that are close. If this is the case, then:

-      how many neighbors are considered?

-      how do you handle the “gender” feature?

Please clarify in the manuscript. To me, this is very interesting and, therefore, a technique that I would like to try myself.

Author response #2: 

how many neighbors are considered? -    5 (default) (Page 8) (Line 316)

how do you handle the “gender” feature?

In oversampling data generated through SMOTE technique with k=5 (default) nearest neighbours. SMOTE select random data from the minority class, then select k-nearest neighbours from the data. Artificially data would then be generated between the random data and the randomly selected k-nearest neighbour. For categorical features smote use different interpolation method such as selects the most common class of the nearest neighbors or different distance metric instead of euclidean distance in the encoded space [67].

The concern of the Reviewer has been executed (Highlighted by yellow color in the manuscript). Thanks for the suggestion. (Page 8) (Line 316- 322)

Explanation

SMOTE itself wouldn't work well for a categorical only feature-set for a few reasons:

  1. It works by interpolating between different points. Depending on how the data is encoded, you might end up with some undefined class (when using one-hot encoding, you might end up with a point that is half of one class and half of another class), or you might end up with a correct class but it doesn't make any sense from an interpolation point of view (for example, if you encode for example the country on a numerical scale like 1 -> US, 2 -> UK, 3 -> NZ, but it doesn't make much sense to interpolate between US and NZ and end up in UK).
  2. SMOTE uses k-means to select points to interpolate between. If you encode your categorical features using one-hot-encoding, you typically end up with a lot of sparse dimensions (dimensions that most points take only the value 0 in). k-means typically won't perform very well in such a space, and points that are nearby in this space might not look a lot like each other.

What you can do is use a modification of the SMOTE algorithm, called SMOTE-N (see https://imbalanced-learn.org/dev/over_sampling.html#smote-variants), which works when all features are categorical. This modifies the SMOTE algorithm to

  1. Use a different interpolation method: selects the most common class of the nearest neighbors
  2. Use a different distance metric (Value Difference Metric) instead of Euclidean distance in the encoded space.

In that link this method is attributed to the original SMOTE paper (https://www3.nd.edu/~dial/publications/chawla2002smote.pdf) where it's found in Section 6.2. There is also SMOTE-NC which is a combination of SMOTE and SMOTE-N for data which has both numerical and categorical features.

For your example, let's say for some reason 3 of the points given: MALE ME RESEARCHER UK default FEMALE BSc Admin staff NZ default
FEMALE MS Scientist sweden default

are considered nearby each other and are used for interpolation. Then a possible added point by SMOTE-N would be:

  • FEMALE (because that's the majority class)
  • MS (all 3 classses have equal frequency, so a class is randomly picked)
  • RESEARCHER (idem to above)
  • NZ (idem to above)
  • default (majority class)

Comment #3:

I noticed that in Table 3, the recall and the sensitivity (SN) are reported at the time. These two metrics are the same; thus, only one of them should be reported.

My main concern here is that the reported recall and SN are different for some ensemble methods:

-       XGBoost: Recall: 0.972; SN: 0.970

-       Balanced random forest: Recall: 0.840; SN: 0.844

-       BalancedBagging: Recall: 0.800; SN: 0.794

-       RUSBoost: Recall: 0.870; SN: 0.979

 I agree that using different performance metrics is essential to have a better picture of the system; however, the tables are so dense, and the discussion in section 4 is so confusing that I suggest using only Accuracy, Sensitivity, Specificity, and AUC (ROC) to analyze and report your results. Given that you only have 2 classes, these four metrics are enough.

Also, do not limit the discussion to the AUC metric. The AUC is a good indicator of the separability of the two classes (PD vs. HC), but it does not tell how well the system classifies the samples. For instance, in Table 3, the AUC for Random Forest is 0.952, the Sensitivity 0.983, and the Specificity 0.622. This means that the classifier is very confident about the correctly classified samples (high AUC), but it diagnoses many of the HC as PD patients (low Specificity).

Author response #3:  Concern # 3 of the Reviewer has been executed.  Thanks for the suggestion. We modify the Table 3 Accuracy, Sensitivity, Specificity, and AUC (ROC), In addition we modify the discussion of Table 3 in section 4.2 (Page 10 ) (Line 412- 426)  (Highlighted by yellow color in the manuscript)

Minor comments #4:

The paper still needs some careful reading. The wording and grammar used in some sections make the manuscript very hard to read. Here are some examples, but I cannot do it for the whole document because it is simply too much work.

-       The abbreviation PD should also be defined in line 34. Additionally, the use of “PD” is very random in the current version of the document.

-       Line 99: Imbalanced -> imbalanced

-       Line 425: while->While      

-       Line 281-282: “Furthermore, the vowel letter "a" pronounced by the PD patient was collected in replicates of three” -> All participants were asked to perform the sustained phonation of the vowel “ah” three times.

-       Line 282: Speech effect is -> Speech disorders are.

-       Figure 1 is too small. Zooming-in does not help because the figure has a low resolution.

Author response:  Concern # 4 of the Reviewer has been executed. Thanks for the suggestion.

Reviewer 4 Report

My concerns have been addressed.

Author Response

thank you

Round 3

Reviewer 1 Report

The authors' contribution is still not enough the paper to be accepted. The authors use and combine well-known methods and tools to get results. it's looks like evaluation report and can be ucceptable for a conference but it is not suitable for the journal. 

Moreover, the results of the paper are not clear because the authors use only one dataset. It is not enough.

Author Response

Our contributions are

  1. We considered an Imbalanced dataset and performed automatic classification between PD patients and healthy controls to evaluate the robustness of different ensemble methods for class Imbalance.
  2. Decision tree ensembles have been shown to have excellent performance in different domains. In this study, we carried out extensive performance evaluations of different types of decision tree ensembles such as RUSBoost, Isolation Forset, RUSBagging, Balanced Bagging etc. Those are developed for imbalanced data. To the best of our knowledge which never been used by other researcher in this area.
  3. We carried out the feature selection using Lasso and Information gain method to achieve the best set of features.

The detailed study considering the data as imbalanced study has not been done. Therefore, we believe that our contributions add value to this area. As the dataset has been used by many researchers as benchmark dataset for their research (please refer literature survey section), we also did the experiment with this dataset.

Reviewer 3 Report

Thanks to the author for taking the time to address my comments. 

I don't have any other major concerns.

Author Response

thank you
